# Dietary Polyphenol Intake and Gastric Cancer: A Systematic Review and Meta-Analysis

**DOI:** 10.3390/cancers14235878

**Published:** 2022-11-29

**Authors:** Marcela de Araújo Fagundes, Alex Richard Costa Silva, Gisele Aparecida Fernandes, Maria Paula Curado

**Affiliations:** 1Graduate Program of A.C. Camargo Cancer Center, R. Professor Antônio Prudente, 211, São Paulo 01509-900 , Brazil; 2Group of Epidemiology and Statistics on Cancer, International Research Center, A.C. Camargo Cancer Center, R. Tagua 440, Sao Paulo 01508-010, Brazil

**Keywords:** dietary polyphenol intake, gastric cancer, systematic review, meta-analysis

## Abstract

**Simple Summary:**

Phenolic compounds are the most abundant antioxidants in the human diet. There are about eight thousand types of phenols, but the main classes are flavonoids, phenolic acids, stilbenes, lignans, and other polyphenols. These compounds have anticarcinogenic properties and may exert a protective effect against several cancers, including Gastric Cancer (GC). However, previous studies investigating GC have focused on flavonoids and the results are controversial. Our systematic review with meta-analysis is the first to report the association between total polyphenol intake, as well as the consumption of more than two classes of polyphenols concomitantly, and GC risk. Polyphenol consumption decreased the risk of GC in both sexes, but to a greater extent in females. The risk reduction was greater in studies carried out in Europe and Asia, therefore further studies in the Latin American population are warranted. Considering the high incidence rates of GC worldwide and the fact modifiable risk factors, such as diet, are amenable to intervention, the findings from this systematic review and meta-analysis can help strengthen strategies to encourage the consumption of foods rich in polyphenols and reduce the incidence of GC.

**Abstract:**

Background: Phenolic compound consumption may have a protective effect against gastric cancer (GC). Most GC studies focus on the flavonoids class, but results are conflicting and knowledge gaps remain for other classes and total polyphenol intake. This study aimed to assess the association between polyphenol intake (total, flavonoids, and other classes) and GC. Methods: In this systematic review and meta-analysis, the PubMed, Embase, Scopus, LILACS, Web of Science, and OpenGrey databases were searched for studies published up to 20 March 2022. Case–control and cohort studies analyzing the association between polyphenol intake and GC were included. For the meta-analysis, pooled summary estimates were calculated using a random-effects model, and the estimates extracted adjusted for most variables. Subgroup analyses were performed for subclass (e.g., flavonoids and other classes), sex, geographical area, study design, anatomical subtype, histological subtype, family history of GC and fruit and/or vegetable intake. The study was registered with PROSPERO (#CRD42022306014). Findings: The search identified 2752 records, of which 19 studies published during the period 1999–2021 including a total of 1,197,857 subjects were eligible. Polyphenol consumption reduced GC risk by 29% (RR = 0.71; 95% CI: 0.62–0.81; I^2^ = 60.5%); while flavonoid intake decreased GC risk by 28% (RR = 0.72; 95% CI: 0.61–0.85; I^2^ = 64.3%), similar to the reduction fort other classes (RR = 0.65; 95% CI: 0.54–0.79; I^2^ = 72.0%). Protective effects against GC were observed in both sexes (male, RR = 0.79; 95% CI: 0.67–0.94, I^2^ = 31.6%; female, RR = 0.65; 95% CI: 0.48–0.87, I^2^ = 49.7%) and for intestinal subtype (RR = 0.65; 95% CI: 0.52–0.82, I^2^ = 0.0%). By continent, polyphenol consumption reduced GC risk in both Europe (RR = 0.67; 95% CI: 0.57–0.79, I^2^ = 44.2%) and Asia (RR = 0.67; 95% CI: 0.51–0.89, I^2^ = 60.7%). Conclusions: Dietary polyphenol intake decreased GC risk. The reduction was greatest in females. Most previous studies were carried out in Europe and Asia. Further studies investigating polyphenol consumption and GC in Latin American populations are warranted.

## 1. Research in Context

### 1.1. Evidence before Study

Polyphenols have anticarcinogenic, anti-inflammatory, and antimicrobial properties and act on cellular proliferation pathways to inhibit the growth of *H. pylori*. Therefore, the ingestion of polyphenols might be associated with the inhibition of cancers of the digestive tract. Since 1999, studies have suggested that flavonoids exert a protective effect against gastric cancer, but results are inconsistent. There are about 8000 phenols that can be categorized into various classes and subclasses, with flavonoids, phenolic acids, lignans, stilbenes, coumarins, and tannins being the most common. Despite this complexity, most studies have focused on investigating the association of only one of these polyphenol classes (flavonoids), while comprehensive analyses of the effect of total polyphenol intake, as well as its subclasses, on gastric cancer risk are lacking.

### 1.2. Added Value of Study

To the best of our knowledge, this systematic review with meta-analysis is the first to report the association between total polyphenol intake, as well as the consumption of more than two classes of polyphenols concomitantly, and gastric cancer risk. We identified inverse associations between dietary polyphenol intake (total, flavonoids, and other classes including lignans, phenolic acids, stilbenes, and other polyphenols) and gastric cancer. Subgroup analyses by sex, study population, study design, and Lauren histological classification were also performed.

### 1.3. Implications of Available Evidence

Guidelines recommend a high consumption of fruits and vegetables to prevent against cancer. This dietary advice may be related to the presence of polyphenols in these foods, whose ingestion reduces the risk of gastric cancer, regardless of sex. Considering the high incidence rates of gastric cancer worldwide and the fact modifiable risk factors, such as diet, are amenable to intervention, the findings from this systematic review and meta-analysis can help strengthen strategies to encourage the consumption of foods rich in polyphenols and reduce the incidence of gastric cancer.

## 2. Introduction

Gastric cancer (GC) ranks fifth in incidence and fourth in lethality among all cancers. Estimates suggest there were over one million new cases in 2020, representing 5.6% of all cancer cases and around 769,000 deaths [1]. Among the various factors involved in the carcinogenesis of GC, epidemiological studies have suggested that a high intake of fruit and vegetables is inversely associated with the risk of GC [2,3]. This reduced risk is attributed to the intake of nutrients, micronutrients, and other dietary compounds, including antioxidants such as polyphenols [4,5].

Polyphenols have shown specific and nonspecific anticarcinogenic properties and may exert a protective effect against several cancers, including GC [6]. These compounds have diverse chemical structures, with the most common variations in the chemical skeleton including the degree of oxidation, hydroxylation, methylation, and glycosylation. The main classes of polyphenols are flavonoids, phenolic acids, stilbenes, lignans, and other polyphenols [5]. Stilbenes, lignans, and tannins are classified as nonflavonoids, and some subclasses are shown in Figure 1.

Previous studies investigating GC have focused on specific polyphenol classes [8,9,10,11]. Flavonoids are the most-studied polyphenol class [10,11,12,13]. An inverse association between flavonoid consumption and GC has been observed in Greece [9], Italy [14], and Korea [13]. Similarly, multicenter case–control studies carried out in Europe and the USA have found that individuals who consume greater amounts of polyphenols have a lower risk of developing GC [4,5,15]. However, cohort studies conducted in Finland [12,16] and Japan [10] failed to demonstrate an association between polyphenol intake and GC. Moreover, most of the previously published meta-analyses on this topic only analyzed the intake of a single class of polyphenols (flavonoids). In addition, the results of subanalyses proved inconsistent, possibly due to the small number of studies assessing the variables of interest in the subanalysis [17,18]. Therefore, we performed a systematic review and meta-analysis investigating the association between GC risk and dietary polyphenol intake, for the main classes and subclasses of polyphenols.

## 3. Methods

### 3.1. Search Strategy and Selection Criteria

This systematic review and meta-analysis followed the PRISMA guidelines for the reporting of meta-analyses [19]. The electronic databases PubMed, Scopus, Embase, LILACS, OpenGrey, and Web of Science were searched for studies published from the date of each database’s inception up to 20 March 2022 that assessed the association between GC risk and dietary polyphenol intake, using the following search terms: (polyphenol or provinols or phenol or flavonoid or flavonol or “phenolic acids” or “hydroxycinnamic acids” or “hydroxybenzoic acids” or “hydroxyphenylacetic acids” or lignans or stilbenes or alkylphenols or alkylmethoxyphenols) and (“gastric cancer” or “gastric tumor” or “stomach cancer” or “gastric adenocarcinoma” or “neoplasm, stomach” or “stomach neoplasm”). The full search strategies used for each database are described in Appendix A. A manual search for additional citations was also performed by examining the reference lists of the articles retrieved. Articles were included if they: assessed the association between the intake of dietary polyphenols and risk of GC; were observational studies, such as cohort or case–control studies, with no language restrictions; and polyphenol intake was measured by validated questionnaires. Articles in languages other than English were translated if necessary. Two authors (MAF and ARCS) independently screened the title and abstract of potentially eligible articles according to the eligibility criteria, and any duplicates were excluded. The researchers performed blind double-checks, and areas of disagreement or uncertainty were resolved by consensus among all authors. When the eligibility criteria were met based on the title and abstract screening, the full text was retrieved for data extraction. The following types of articles were excluded: (1) cell studies and animal studies; (2) review articles, letters to the editor, case reports, ecological studies, and cross-sectional studies; and (3) studies in which polyphenol levels were measured in blood or urine.

### 3.2. Data Analysis

Two authors (MAF and ARCS) independently extracted the following information from all eligible studies: (1) last name of the first author; (2) year of publication; (3) country; (4) study design; (5) period of study; (6) total number of participants, numbers of cases and controls; (7) polyphenol assessment method; (8) food source of polyphenols; (9) classes of polyphenols studied; (10) relative risk (RR)/odds ratio (OR) and 95% confidence interval (CI) indicating the highest versus the lowest categories of dietary polyphenol intake where estimates extracted were adjusted for most variables; (11) intake comparison; and (12) adjustment variables (family history of GC and fruit and/or vegetable intake).

The most adjusted OR (highest versus lowest intake of polyphenols) was considered to be an approximation of the RR, and the summary results were reported as RR for simplicity. Random-effects meta-analysis was used to assess the association between dietary polyphenol intake, including the main subclasses (exposure), and GC risk (outcome), with pooled ORs calculated together with their 95% CIs.

Cochran’s chi-squared test and the I^2^ test were used to quantify the heterogeneity of the studies where, for *p* < 0.05 and/or I^2^ > 50%, substantial heterogeneity existed among studies [20,21]. The choice of ORs for analysis was made considering all classes of polyphenols. The order of choice of ORs was: total polyphenols, total flavonoids, flavanones, phenolic acids, lignans and stilbenes. The ORs for both sexes were selected. When studies were analyzed by sex and anatomical type separately, female gender and non-cardia type were selected. Subgroup analyses by subclasses (flavonoids and non-flavonoids), sex (male or female), study design (case–control or cohort study), topography (cardia or noncardia), histological subtype (diffuse or intestinal), geographical area (Europe, Asia or America), and adjustment variable (family history of GC and fruit and/or vegetable intake) were also performed to identify potential confounders/modifiers. Adjusted risk estimates in subgroup analyses were for the specific variables cited. Regarding study design and geographic area, risk estimates were for both sexes and chosen in the order outlined above for general meta-analysis. For studies providing fully adjusted results (multiple variables), ORs were extracted from statistical models that provided adjustment for the variable of interest, in the case of subgroup analyses by adjustment variable, such as family history of gastric cancer or fruit and vegetable intake.

The Newcastle–Ottawa Scale [22] was used to assess the risk of bias by determining the quality of the observational studies selected using two independent scales (for cohort and case–control studies). The scale consists of items divided into three domains: selection, comparison and exposure (case–control studies) or outcome (cohort studies). Studies with a rating of 6 or higher were considered high quality [23]. Egger’s test and funnel plots were conducted to assess potential publication bias, where a value of *p* < 0.05 was considered a statistically significant difference for all tests used. All statistical analyses were conducted using the software STATA, version 15.0 (Stata Corporation, College Station, TX, USA). This study was registered with PROSPERO (#CRD42022306014).

## 4. Results

The literature search led to the retrieval of 2752 records, of which 637 were excluded as duplicates and a further 2096 excluded after initial screening of titles and abstracts. Two additional articles were later identified in the references of the screened articles [16,24]. A total of 21 full-text articles were assessed for eligibility (Figure 2), with subsequent exclusion of two articles because polyphenol content was measured in plasma, giving 19 studies for inclusion in the review. All studies, published during the period 1999–2021 and involving 1,197,857 participants (patients and controls), were included in the meta-analysis. The median Newcastle–Ottawa rating for the 19 studies reviewed was 7 (range: 6–9; Appendix A). Quality score ranges were 8–9 for the cohort studies and 6–8 for the case–control studies. Therefore, all studies were considered to be of high quality.

The main characteristics of the included studies are summarized in Table 1; seven were cohort studies, and 12 were case–control studies. Eleven studies were conducted in Europe (multiple European countries, *n* = 2; Italy, *n* = 1; Greece, *n* = 1; Sweden, *n* = 2; Finland, *n* = 2; and Spain, *n* = 3), five studies were conducted in Asia (Korea, *n* = 3; Japan, *n* = 2), and only 3 studies were conducted in North America (USA, *n* = 2; Mexico, *n* = 1). Over half the studies were published after 2010 (*n* = 12). Twelve studies analyzed the flavonoid class only, whereas two investigated 3 classes of polyphenols. A total of 16 studies used food-frequency questionnaires to measure polyphenol intake. The confounders used by the studies were: age, sex, education, total energy intake, body mass index, physical activity, smoking, alcohol intake, and total fruit and vegetable consumption. All variables for each statistical model are shown in Table 1.

The pooled RRs with 95%CIs (highest versus lowest categories of dietary polyphenol intake) were calculated to assess the association between dietary polyphenol intake and risk of GC. As shown in Figure 3, consumption of polyphenols reduced the risk of GC by 29% (RR = 0.71; 95% CI: 0.62–0.81; I^2^ = 60.5%). The analysis of flavonoid consumption and risk of GC showed that the intake of total flavonoids or some types of flavonoids had an inverse association with GC (RR = 0.72; 95% CI: 0.61–0.85; I^2^ = 64.3%) (Figure 4). Likewise, there was an association between the consumption of other classes of polyphenols (lignans, phenolic acids, and other polyphenols) and GC. As shown in Figure 5, the consumption of other classes of polyphenols reduced the risk of GC by 35% (RR = 0.65; 95% CI: 0.54–0.79; I^2^ = 72.0%). Other data combinations were also made to assess the effects of sex (male or female) and anatomical subtype (cardia or noncardia), and the meta-analysis showed a significant inverse association between polyphenol intake and GC risk (Appendix A). Heterogeneity was substantial for most of the pooled estimates. The funnel plot shows no evidence of asymmetry (Figure 6; Appendix A), and Egger’s test showed that no publication bias was detected in this meta-analysis (*p* = 0.316).

On subgroup analyses, the protective effects of polyphenol intake against GC were observed in both sexes (male, RR = 0.79; 95% CI: 0.67–0.94, 8 studies, P for heterogeneity = 0.176, I^2^ = 31.6%; female, RR = 0.65; 95% CI: 0.48–0.87, 7 studies, P for heterogeneity = 0.064, I^2^ = 49.7%) and intestinal histological type (RR = 0.65; 95% CI: 0.52–0.82, 3 studies, P for heterogeneity = 0.494, I^2^ = 0.0%). When the analysis was stratified by geographical area, an inverse association between polyphenol intake and GC was observed in both Europe (RR = 0.67; 95% CI: 0.57–0.79, 11 studies, P for heterogeneity = 0.056, I^2^ = 44.2%) and Asia (RR = 0.67; 95% CI: 0.51–0.89, 5 studies, P for heterogeneity = 0.038, I^2^ = 60.7%). Regardless of whether the studies were adjusted for a family history of GC or fruit/vegetable intake, the protective effect of polyphenol intake against GC was observed (Table 2).

## 5. Discussion

This systematic review and meta-analysis showed an inverse association between polyphenol consumption and GC, for the intake of total polyphenols and of the main classes, including flavonoids, lignans, phenolic acids, stilbenes, and other polyphenols. Previous studies report conflicting results regarding the relationship between polyphenol intake and GC risk. Multicenter case–control studies carried out in Europe and the USA have found that individuals who consume greater amounts of polyphenols have a lower risk of developing GC [4,5,15,31]. However, cohort studies carried out in Finland [12,16], Japan [10] and the USA [11] failed to demonstrate this association. In the present systematic review, there was a dearth of studies analyzing the consumption of polyphenols and GC risk in Latin American countries, with a sole investigation in Mexico [25].

On the current subgroup analysis by geographic location, an inverse association between polyphenol intake and GC was identified in Europe and Asia but not in North America. The results of other meta-analyses also indicate an association in European populations only [17,33], where one of the reasons for this phenomenon may be the diversity of the diets across the different studies. The level of polyphenols contained in vegetables and fruit, for example, depends on the type of cultivation, crop variety and location, as well as the specific morphology of the plant source. In addition, ethnic differences in food structure or cultural differences in the storage and preparation of foods, particularly those of vegetables might also account for this result [34,35]. Other factors which might explain the geographic disparities found, beyond polyphenols, include the fact that the populations surveyed differ for other characteristics, as do the studies reviewed.

Flavonoids are the most-studied polyphenol class included in systematic reviews and meta-analysis studies [6,13,17,30,33,35,36,37,38,39]. In the meta-analysis performed by Bo et al. [37], no association was found between the highest dietary flavonoid intake and the risk of digestive tract cancers, including GC. In contrast, the meta-analyses of GC risk and dietary flavonoids by Xie, Huang, and Su [33], and by Woo and Kim [18], showed an association between flavonols and GC risk based on a limited number of selected studies. In the study by Woo and Kim [18], several subclasses of flavonoids, mostly in case–control studies, showed protective effects against stomach cancer risk, but total dietary flavonoid intake was not associated with a reduced risk of stomach cancer. Likewise, Grosso et al. [38], in a meta-analysis on dietary flavonoid and lignan intake with cancer risk, found no association between lignan intake and GC. This result differs from the findings of the present study, which showed lower risk of GC with increased flavonoid consumption. In addition, intake of lignans, phenolic acids, stilbenes, and other polyphenols was associated with a reduced risk of GC. Within natural compounds, polyphenols (including all classes and subclasses) represent a large diverse group used in the prevention and treatment of cancer [39]. Numerous studies have demonstrated that dietary polyphenol compounds exhibit a variety of bioactivities that can repress carcinogenesis and cancer progression, such as anticarcinogenic, anti-inflammatory, and antimicrobial properties, and act on cell proliferation pathways to inhibit the growth of *H. pylori* [36,38,40,41,42]. Furthermore, in gastric cancer, the immunological paradigm must also be taken into account, given reports suggesting some foods rich in polyphenols may improve immune function of gastric cells, besides improving the immune functions of lymphocyte proliferation, cellular and humoral immunity responses, thymocyte differentiation, and tumor immunity [38,43,44].

In the present study, subgroup analysis by study design was also conducted, revealing lower GC risk in case–control studies but not for cohort studies. Bo et al. [17] and You et al. [21] found that neither case–control nor cohort studies showed an association between polyphenol intake and GC risk. In the present analysis, heterogeneity was higher among the cohort studies, whereas the presence of greater recall and selection bias in case–control studies may potentially have led to a spurious association.

The analysis of the association between polyphenol intake and GC risk by sex revealed a significant risk reduction for both men and women. This finding differs from the results observed by other meta-analyses, in which either no association was found for males and females [37] or an inverse association was seen for women only [35]. In the present study, the risk reduction was greater for females, which may be partly explained by the fact that polyphenols can regulate female hormones and play a protective role against cancer. Similarly, a large prospective study has demonstrated associations between stomach cancer risk and age at menopause, years of fertility, and years since menopause [45]. However, to the best of our knowledge, this is the first meta-analysis to identify a lower risk of GC for males, suggesting that a diet rich in polyphenols has preventive effects against GC for both sexes.

However, no association was found between polyphenol intake and reduced risk of noncardia gastric adenocarcinoma or gastric cardia adenocarcinoma. These findings are similar to the results of Yang et al. [24]. In contrast, a study by Xie, Huang, and Su [33] found an association between flavonol intake and reduced risk of noncardia gastric adenocarcinoma only. Regarding histological subtype, the present study showed a reduction in risk for intestinal-type GC but not for the diffuse type. A study published by Ekström et al. [46] found that a high intake of antioxidants, as a consequence of a high consumption of fruit and vegetables, was associated with a decreased risk of cardia and noncardia GC of both the intestinal and diffuse types. Few studies have analyzed the association between polyphenol intake and the histological subtypes of GC [26,27,31], where the present meta-analysis is the first investigation to assess these pooled results. However, the subanalysis by histological type was based on 3 studies and results should therefore be interpreted with caution.

The subgroup analyses also revealed an association between polyphenol intake and GC risk, irrespective of adjustments for family history of cancer or fruit and vegetable intake. Polyphenols are considered mediators of the protective effects of vegetables and fruit against various forms of cancer and chronic diseases. However, as these foods contain other compounds, it is important to identify which compound mediates the preventive potential of fruit and vegetables. To this end, some studies have used fruit and vegetable intake in statistical models to assess whether the association between polyphenol intake and GC remains significant [9,10,15]. Given that 7 out of the 19 studies included in the present review used fruit and vegetable intake as an adjustment factor, this subanalysis was performed, revealing that the risk of GC decreased independently of adjusting for fruit and vegetable intake. The subanalysis also served to determine whether family history of GC was a potential confounder, confirming that the association between polyphenol intake and GC was independent of this adjustment.

The present study has several strengths. First, a large sample drawing on a satisfactory number of studies was analyzed. A total of 19 studies were included, allowing a subgroup analyses to be carried out. Limitations of the meta-analysis included that some studies stratified results by sex and anatomical localization, while most investigations on the topic addressed only one class or subclass of polyphenols, hampering comparison with other studies.

## 6. Conclusions

In conclusions, the current study provided evidence for an inverse association between dietary polyphenol intake and GC risk, for total polyphenol intake as well as for various polyphenol classes, in a representative sample. Polyphenol consumption decreased the risk of GC in both sexes, but to a greater extent in females. Moreover, the risk reduction was greater in studies carried out in Europe and Asia. Further studies investigating the association of polyphenol consumption with GC in the Latin American population are warranted.

## Figures and Tables

**Figure 1 cancers-14-05878-f001:**
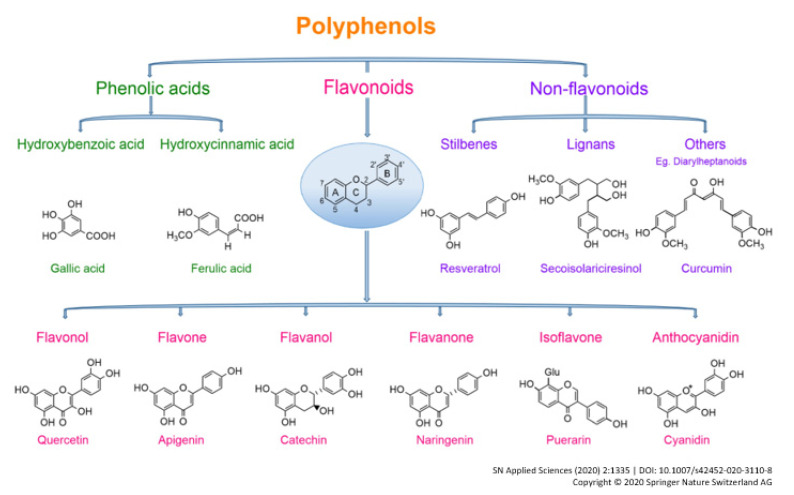
Classification of polyphenols [7].

**Figure 2 cancers-14-05878-f002:**
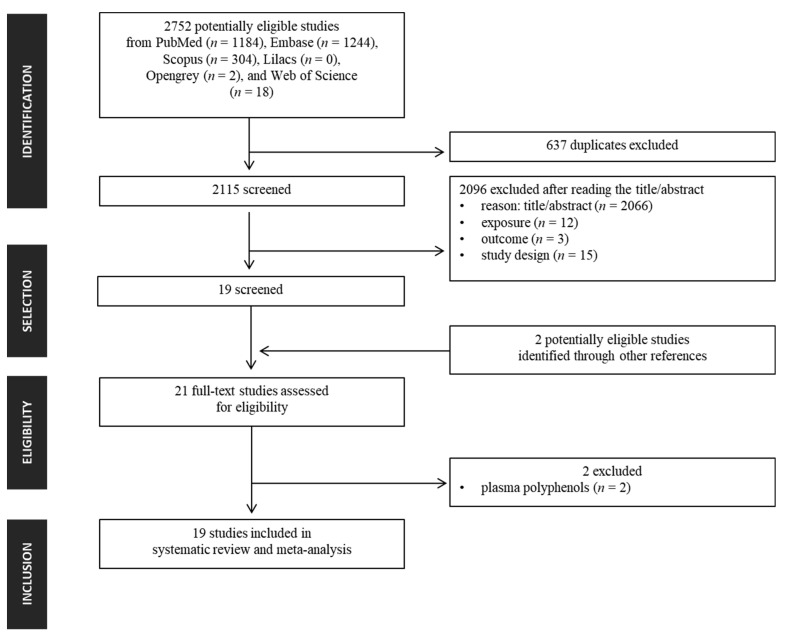
Flow diagram of selection of eligible studies for inclusion in systematic review according to PRISMA guidelines.

**Figure 3 cancers-14-05878-f003:**
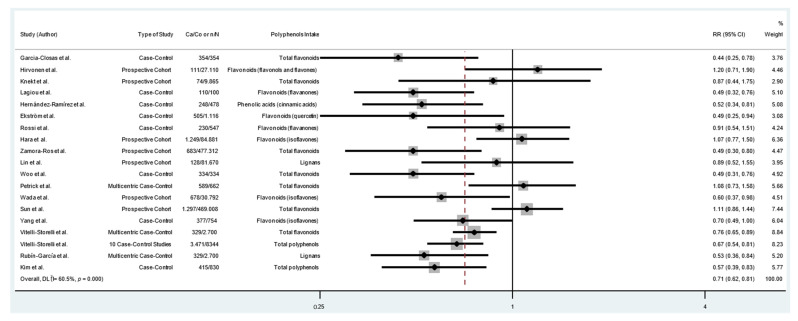
Forest plot of polyphenol consumption and gastric cancer risk. Case–control and cohort studies are presented in chronological order of publication. Square boxes represent study-specific estimates. Markers vary in size according to weight assigned to each study, the size of each box reflects the study’s weight in the analysis, and horizontal lines represent 95% confidence intervals. Diamonds show the pooled effect. Abbreviations: RR, relative risk; Ca, number of cases; Co, number of controls; n, number of events; N, number of patients [4,5,8,9,10,11,12,13,14,15,16,25,26,27,28,29,30,31,32].

**Figure 4 cancers-14-05878-f004:**
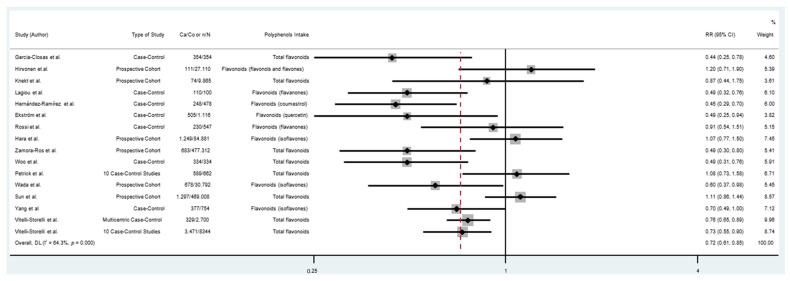
Forest plot of flavonoid consumption and gastric cancer risk. Case–control and cohort studies are presented in chronological order of publication. Square boxes represent study-specific estimates. Markers vary in size according to weight assigned to each study, size of each box reflects the study’s weight in the analysis, and horizontal lines represent 95% confidence intervals. Diamonds show the pooled effect. Abbreviations: RR, relative risk; Ca, number of cases; Co, number of controls; n, number of events; N, number of patients [4,5,8,9,10,11,12,13,14,16,25,26,27,29,30,31].

**Figure 5 cancers-14-05878-f005:**
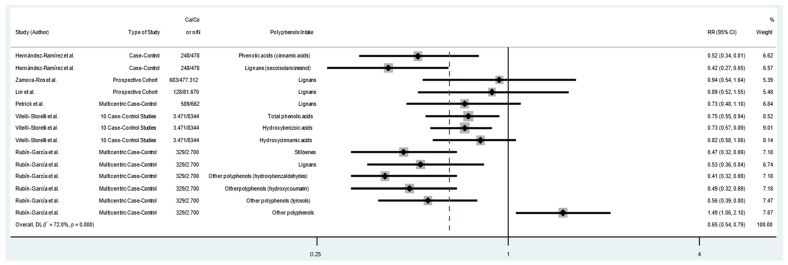
Forest plot of consumption of lignans, phenolic acids, stilbenes, and other polyphenols and gastric cancer risk. Case–control and cohort studies are presented in chronological order of publication. Square boxes represent study-specific estimates. Markers vary in size according to weight assigned to each study, size of each box reflects the study’s weight in the analysis, and horizontal lines represent 95% confidence intervals. Diamonds show the pooled effect. Abbreviations: RR, relative risk; Ca, number of cases; Co, number of controls; n, number of events; N, number of patients [4,5,15,1,25,27,28].

**Figure 6 cancers-14-05878-f006:**
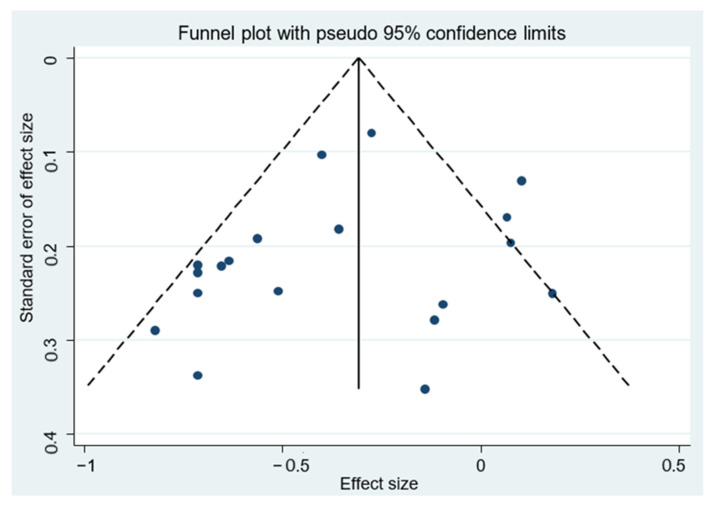
Funnel plot of dietary polyphenol intake and gastric cancer risk.

**Table 1 cancers-14-05878-t001:** Characteristics of studies on dietary polyphenol intake and gastric cancer risk.

Author/Country	Study Type and Period	Sample	Diet Assessment Method	Source of Polyphenols	Classes of Polyphenols	RR/OR (95% CI)	Intake Comparison	Adjustment Variables
Garcia-Closas et al. [8] Spain	Case-control 1987–1989	Cases = 354 Controls = 354	Past year’s diet history	Fruit, vegetables, fruit juices, wines and tea infusions	Total Flavonoids Quercetin Kaempferol Myricetin	0.44 (0.25–0.78) 0.62 (0.35–1.10) 0.48 (0.26–0.88) 1.12 (0.67–1.85)	Highest quartile X Lowest quartile	Intake of nitrites, nitrosamines, vitamin C, total energy, total carotenoids, and other specific flavonoids.
Hirvonen et al. [12] Finland	Prospective cohort 1985–1993	27,110 GC = 111	Past year’s diet history	Fruit, vegetables, teas, wines and sweets	Flavonoids and Flavones	1.2 (0.71–1.9)	Highest quartile X Lowest quartile	Age and supplementation group.
Knekt et al. [16] Finland	Prospective cohort 1966–1972	9865 GC = 74	Past year’s diet history	Fruit, vegetables, sweets and beverages (including tea and wines)	Total Flavonoids Quercetin Kaempferol Myricetin Hesperetin Naringenin	0.87 (0.44–1.75) 1.03 (0.52–2.07) 1.14 (0.59–2.22) 1.16 (0.59–2.26) 0.88 (0.43–1.80) 0.94 (0.47–1.88)	Highest quartile X Lowest quartile	Sex, age, geographic area, occupation, smoking, and BMI.
Lagiou et al. [9] Greece	Case-control 1981–1984	Cases = 110 Controls = 100	FFQ last 5 years	Items not described		Model 1/2	Per one standard deviation increment:	Model 1: Age, gender, place of birth, BMI, height, years of education, smoking habits and duration of smoking, alcohol consumption, total energy intake. Model 2: Model 1 + fruit and vegetable consumption.
					Flavanones	0.49 (0.32–0.76)/0.55 (0.31–0.96)	per 19.8 mg/day	
					Flavan-3-ols	1.10 (0.76–1.60)/1.04 (0.68–1.58)	per 135.1 mg/day	
					Flavonols	0.40 (0.25–0.64)/0.77 (0.42–1.40)	per 10.0 mg/day	
					Flavones	0.60 (0.40–0.89)/0.70 (0.43–1.14)	per 0.3 mg/day	
					Anthocyanidins	0.88 (0.60–1.28)/1.14 (0.72–1.80)	per 40.4 mg/day	
					Isoflavones	1.27 (0.84–1.93)/1.16 (0.73–1.84)	per 2.0 mg/day	
Hernández-Ramírez et al. [25] Mexico	Case control 2004–2005	Cases = 248 Controls = 478	FFQ last 3 years	Fruit, vegetables, noodle soup, hot sauce, beans, orange juice, red wine		Model 1/2/3/4	Highest tertile X Lowest tertile	Model 1: Energy, age, gender, *H. pylori* CagA status, schooling and consumption of salt, chili, alcohol. Model 2: Model 1 + vitamins C and E. Model 3: Model 2 + fruits and vegetables. Model 4: Model 3 + polyphenols.
					Cinnamic Acids	0.52 (0.34–0.81)/0.49 (0.31–0.78)/ 0.61 (0.38–0.97)/0.80 (0.49–1.31).		
					Secoisolariciresinol	0.42 (0.27–0.65)/0.41 (0.26–0.64)/0.47 (0.30–0.74)/0.57 (0.32–0.99).		
					Coumestrol	0.45 (0.29–0.70)/0.45 (0.29–0.71)/0.42 (0.27–0.65)/0.67 (0.39–1.16).		
Ekström et al. [26] Sweden	Case control 1989–1995	Cases = 505 Controls = 1116	FFQ last 20 years	Fruit, vegetables, wine, tea, coffee and fruit juices.	Quercetin	Male: 0.66 (0.42–1.04) Female: 0.49 (0.25–0.94) Cardia: 0.76 (0.40–1.44) Noncardia: 0.57 (0.40–0.83) Intestinal: 0.51 (0.32–0.82) Diffuse: 0.54 (0.31–0.92)	Highest quintile X Lowest quintile	Age, gender, socioeconomic status, number of siblings, body mass index, smoking, and energy and salt intake.
Rossi et al. [14] Italy	Case control 1997–2007	Cases = 230 Controls = 547	FFQ last 2 years	78 items such as fruit, vegetables, soup, tea, wine and chocolate.	Isoflavones	0.88 (0.53–1.46)	Highest quintile X Lowest quintile	Sex, age, education, year of interview, body mass index, tobacco smoking, and total energy intake.
					Anthocyanidins	0.91 (0.56–1.47)		
					Flavan-3-ols	0.75 (0.45–1.23)		
					Flavanones	0.91 (0.54–1.51)		
					Flavones	0.83 (0.50–1.39)		
					Flavonols	0.62 (0.38–1.02)		
					Proanthocyanidins	0.34 (0.20–0.58)		
Hara et al. [10] Japan	Prospective cohort 1990–2006	84,881 GC = 1249	FFQ past year	Miso soup, food and soy milk	Isoflavones	Model 1/2 Male 0.98 (0.80, 1.20)/1.00 (0.81, 1.24) Female 0.99 (0.71, 1.37)/1.07 (0.77, 1.50) Model 2 only Cardia: 2.00 (0.97, 4.12) Noncardia: 0.97 (0.74, 1.26)	Highest quartile X Lowest quartile	Model 1: Age and public center area. Model 2: BMI, smoking status, ethanol intake, family history of gastric cancer, vegetable intake, fruit intake, fish intake, salt intake, and total energy intake.
Zamora-Ros et al. [27] EPIC study (Denmark, France, Germany, Greece, Italy, Netherlands, Norway, Spain, Sweden and UK)	Prospective cohort 1992–2010	477,312 GC = 683	Several validated FFQs	Items not described	Total flavonoids	Male: 0.97 (0.67, 1.41) Female: 0.49 (0.30, 0.80) Cardia: 0.84 (0.64, 1.11) Noncardia: 0.85 (0.70, 1.03) Intestinal: 0.70 (0.52, 0.94) Diffuse: 0.94 (0.74, 1.19)	Male/Female: Highest quartile X Lowest quartile Cardia/Noncardia; Intestinal/Diffuse: log_2_	Age, educational level, physical activity, BMI, alcohol and energy intake, and daily consumption of fruit, vegetables, and red and processed meat.
					Anthocyanidins	Male: 0.98 (0.68, 1.41) Female: 0.71 (0.44, 1.16) Cardia: 0.89 (0.69, 1.15) Noncardia: 0.90 (0.79, 1.04) Intestinal: 0.92 (0.73, 1.16) Diffuse: 0.86 (0.75, 0.99)		
					Flavonols	Male: 0.93 (0.63, 1.37) Female: 0.45 (0.27, 0.75) Cardia: 0.85 (0.60, 1.20) Noncardia: 0.90 (0.71, 1.13) Intestinal: 0.72 (0.49, 1.06) Diffuse: 1.04 (0.78, 1.40)		
					Flavanones	Male: 0.91 (0.64, 1.28) Female: 1.01 (0.68, 1.50) Cardia: 0.92 (0.81, 1.05) Noncardia: 0.99 (0.90, 1.09) Intestinal: 1.10 (0.92, 1.32) Diffuse: 0.96 (0.87, 1.07)		
					Flavones	Male: 0.86 (0.60, 1.23) Female: 0.59 (0.38, 0.93) Cardia: 0.83 (0.65, 1.07) Noncardia: 0.94 (0.82, 1.07) Intestinal: 0.97 (0.76, 1.24) Diffuse: 0.92 (0.78, 1.09)		
					Flavanols	Male: 0.93 (0.64, 1.34) Female: 0.52 (0.32, 0.83) Cardia: 0.89 (0.70, 1.11) Noncardia: 0.90 (0.78, 1.05) Intestinal: 0.78 (0.65, 0.94) Diffuse: 0.98 (0.81, 1.19)		
					Flavan-3-ol monomers	Male: 0.98 (0.68, 1.40) Female: 0.55 (0.34, 0.88) Cardia: 0.91 (0.79, 1.05) Noncardia: 0.93 (0.84, 1.02) Intestinal: 0.83 (0.72, 0.95) Diffuse: 1.01 (0.89, 1.14)		
					Proanthocyanidins	Male: 0.84 (0.55, 1.27) Female: 0.71 (0.44, 1.15) Cardia: 1.05 (0.71, 1.56) Noncardia: 0.92 (0.78, 1.09) Intestinal: 0.86 (0.71, 1.04) Diffuse: 0.98 (0.78, 1.24)		
					Theaflavins	Male: 1.06 (0.73, 1.54) Female: 0.57 (0.36, 0.91) Cardia: 0.98 (0.95, 1.02) Noncardia: 0.99 (0.97, 1.01) Intestinal: 0.95 (0.92, 0.98) Diffuse:1.02 (0.99, 1.05)		
					Isoflavones	Male: 0.77 (0.50, 1.18) Female: 1.05 (0.61, 1.82) Cardia: 1.13 (0.85, 1.50) Noncardia: 1.00 (0.83, 1.19) Intestinal: 1.08 (0.80, 1.47) Diffuse: 0.90 (0.71, 1.13)		
					Lignans	Male: 0.99 (0.63, 1.55) Female: 0.94 (0.54, 1.64) Cardia: 0.61 (0.33, 1.13) Noncardia: 0.85 (0.59, 1.23) Intestinal: 1.17 (0.65, 2.12) Diffuse: 0.79 (0.50, 1.25)		
Lin et al. [28] Sweden	Prospective cohort 1987–2009	81.670 GC = 128	FFQ	65 unspecified items	Lignans	Model 1/2 0.78 (0.48–1.28)/0.89 (0.52–1.55) Men only 0.81 (0.43–1.55)/0.83 (0.40–1.76)	Highest quartile X Lowest quartile	Model 1: Sex, age, and energy intake. Model 2: Model 1 + education, BMI, alcohol intake, smoking status, gastric ulcer, duodenal ulcer, and diabetes. P.S: For men only, adjustment for sex was not included.
Woo et al. [13] Korea	Case-control 2011–2014	Cases = 334 Controls = 334	FFQ past year	144 unspecified items		Model 1/2	Highest tertile X Lowest tertile	Model 1: Total energy intake, *H. pylori*, age, sex, education, smoking status, alcohol consumption, BMI, physical activity, consumption of pickled vegetables, red and processed meat. Model 2: Model 1 + fruit and vegetable consumption.
					Total flavonoids	Total 0.49 (0.31–0.76)/0.62 (0.36–1.09) Male 0.70 (0.39–1.24)/0.80 (0.39–1.63) Female 0.33 (0.15–0.73)/0.68 (0.25–1.86)		
					Flavonols	Total 0.51 (0.32–0.82)/0.69 (0.39–1.20) Male 0.59 (0.32–1.10)/0.65 (0.32–1.35) Female 0.51 (0.24–1.10)/1.22 (0.47–3.16)		
					Flavones	Total 0.51 (0.31–0.82)/0.72 (0.38–1.35) Male 0.70 (0.38–1.29)/0.84 (0.37–1.89) Female 0.15 (0.06–0.38)/0.22 (0.07–0.67)		
					Flavanones	Total 0.66 (0.43–1.02)/0.92 (0.55–1.52) Male 0.90 (0.52–1.56)/1.12 (0.58–2.17) Female 0.39 (0.18–0.86)/0.64 (0.27–1.52)		
					Flavan-3-ols	Total 0.58 (0.38–0.88)/0.73 (0.45–1.18) Male 0.70 (0.41–1.21)/0.78 (0.41–1.49) Female 0.36 (0.17–0.77)/0.65 (0.27–1.57)		
					Anthocyanidins	Total 0.73 (0.46–1.15)/1.06 (0.62–1.80) Male 0.92 (0.51–1.67)/1.16 (0.57–2.34) Female 0.58 (0.27–1.25)/1.22 (0.49–3.01)		
					Isoflavones	Total 0.72 (0.46–1.12)/0.85 (0.54–1.35) Male 0.90 (0.52–1.54)/0.98 (0.56–1.73) Female 0.51 (0.24–1.08)/0.67 (0.31–1.47)		
Petrick et al. [4] USA	Multi-center Case-control 1993–1995–2000	Cases = 589 Controls = 662	FFQ last 3–5 years	Fruit, vegetables, juices, wine, tea, coffee, pizza, bread, cake, soups, chicken	Total flavonoids	GCA: 1.32 (0.87, 2.00) OGA: 1.08 (0.73, 1.58)	Highest quartile X Lowest quartile	Age, sex, race, geographic center, cigarette smoking, and dietary energy intake.
					Anthocyanidins	GCA: 0.71 (0.46, 1.10) OGA: 0.70 (0.47, 1.03)		
					Flavan-3-ols	GCA: 1.17 (0.77, 1.78) OGA: 1.30 (0.88, 1.92)		
					Flavanones	GCA: 1.23 (0.81, 1.87) OGA: 0.88 (0.60, 1.28)		
					Flavones	GCA: 1.09 (0.71, 1.67) OGA: 1.01 (0.69, 1.50)		
					Flavonols	GCA: 1.42 (0.93, 2.17) OGA: 0.98 (0.67, 1.46)		
					Isoflavones	GCA: 1.56 (0.93, 2.60) OGA: 1.50 (0.96, 2.37)		
					Lignans	GCA: 1.01 (0.65, 1.58) OGA: 0.73 (0.48, 1.11)		
Wada et al. [29] Japan	Prospective cohort 1992–2008	30,792 GC = 678	Diet record-12 days and FFQ (past year)	Miso soup, tofu (soy bean curd), deep-fried tofu, fried tofu, freeze-dried tofu, natto, houba-miso, soymilk and boiled soy beans	Isoflavones	Male: 0.81 (0.60–1.09) Female: 0.60 (0.37–0.98)	Highest quartile X Lowest quartile	Age, BMI, physical activity score, smoking status, alcohol consumption, salt intake and education years for men, and menopausal status for women.
Sun et al. [11] USA	Prospective cohort 1995–2011	469,008 GC = 1297	FFQ past year	116 unspecified items	Total flavonoids	Cardia: 1.02 (0.78, 1.34) Noncardia: 1.11 (0.86, 1.44)	Highest quintile X Lowest quintile	Age, sex, race, education, smoking status, BMI, alcohol intake, self-reported health, vigorous physical activity of ≥20 min and total energy intake.
					Anthocyanidins	Cardia: 1.05 (0.80, 1.39) Noncardia: 0.94 (0.72, 1.23)		
					Flavan-3-ols	Cardia: 1.04 (0.80, 1.36) Noncardia: 1.19 (0.92, 1.54)		
					Flavanones	Cardia: 0.87 (0.68, 1.13) Noncardia: 0.99 (0.76, 1.30)		
					Flavones	Cardia: 0.99 (0.73, 1.34) Noncardia: 1.06 (0.80, 1.40)		
					Flavonols	Cardia: 1.08 (0.80, 1.45) Noncardia: 1.25 (0.94, 1.65)		
					Isoflavones	Cardia: 0.99 (0.73, 1.34) Noncardia: 0.73 (0.54, 0.98)		
Yang et al. [30] Korea	Case control 2011–2014	Cases = 377 Controls = 754	FFQ past year	Legumes, tofu, soymilk, sprouts, and doenjang (Korean traditional fermented soybean paste and soybeans)	Isoflavones	0.70 (0.49–1.00) Male: 0.63 (0.40–0.99) Female: 0.82 (0.45–1.49)	Highest tertile X Lowest tertile	Education, alcohol consumption, smoking status, *Helicobacter pylori* infection, and regular exercise.
Vitelli-Storelli et al. [31] Spain	Multi-center Case-control 2008–2013	Cases = 329 Controls = 2700	FFQ	Vegetables and legumes, fruit, cereals, sweets and snacks, and alcoholic and other beverages.	Total flavonoids	0.76 (0.65, 0.89) Male: 0.51 (0.31–0.82) Female: 0.89 (0.42–1.90) Cardia: 0.67 (0.33, 1.39) Noncardia: 0.55 (0.35, 0.87) Intestinal: 0.74 (0.38, 1.42) Diffuse: 0.38 (0.17, 0.84)	All cases: log2 Male/Female; Cardia/Noncardia; Intestinal/Diffuse: Highest quartile X Lowest quartile	Age, gender, socioeconomic status, area of residence, GC family history, BMI, smoking, physical activity, energy, sodium, red meat, vegetables and past alcohol intake.
					Anthocyanidins	0.88 (0.80, 0.96) Male: 0.47 (0.30–0.74) Female: 1.14 (0.59–2.22) Cardia: 0.62 (0.29, 1.31) Noncardia: 0.67 (0.42, 1.06) Intestinal: 0.61 (0.32, 1.15) Diffuse: 0.92 (0.42, 1.98)		
					Chalcones	0.89 (0.83, 0.95) Male: 0.48 (0.31–0.74) Female: 0.90 (0.80, 1.03) Cardia: 0.40 (0.2, 0.79) Noncardia: 0.60 (0.37, 0.98) Intestinal: 0.39 (0.19, 0.82) Diffuse: 0.89 (0.4, 1.98)		
					Dihydrochalcones	1.02 (0.95, 1.11) Male: 1.35 (0.87–2.09) Female: 0.96 (0.45–2.05) Cardia: 1.92 (0.92, 4.02) Noncardia: 1.38 (0.87, 2.2) Intestinal: 1.4 (0.73, 2.67) Diffuse: 1.23 (0.57, 2.67)		
					Dihydroflavonols	0.89 (0.84, 0.95) Male: 0.38 (0.24–0.59) Female: 0.89 (0.38–2.06) Cardia: 0.60 (0.29, 1.23) Noncardia: 0.47 (0.29, 0.76) Intestinal: 0.54 (0.29, 1.01) Diffuse: 0.49 (0.21, 1.11)		
					Flavan-3-ols	0.82 (0.73, 0.92) Male: 0.49 (0.32–0.77) Female: 0.61 (0.29–1.28) Cardia: 0.65 (0.28, 1.48) Noncardia: 0.59 (0.35, 0.98) Intestinal: 0.57 (0.28, 1.16) Diffuse: 0.42 (0.18, 1.01)		
					Flavanones	0.92 (0.85, 1.00) Male: 0.66 (0.43–1.00 Female: 0.88 (0.42–1.83) Cardia: 0.60 (0.29, 1.23) Noncardia: 0.79 (0.51, 1.22) Intestinal: 0.89 (0.49, 1.64) Diffuse: 0.64 (0.30, 1.36)		
					Flavones	0.99 (0.89, 1.11) Male: 0.75 (0.47–1.20) Female: 1.42 (0.65–3.15) Cardia: 0.70 (0.32, 1.5) Noncardia: 1.27 (0.8, 2.02) Intestinal: 1.36 (0.72, 2.56) Diffuse: 1.64 (0.74, 3.62)		
					Flavonols	0.93 (0.78, 1.10) Male: 0.62 (0.37–1.02) Female: 1.71 (0.74–3.92) Cardia: 0.84 (0.36, 1.98) Noncardia: 1.13 (0.68, 1.88) Intestinal: 1.28 (0.63, 2.56) Diffuse: 1.46 (0.63, 3.4)		
					Isoflavonoids	1.05 (0.98, 1.12) Male: 1.11 (0.65–1.91) Female: 0.99 (0.89, 1.11) Cardia: 1.27 (0.54, 2.98) Noncardia: 1.36 (0.78, 2.38) Intestinal: 1.81 (0.8, 4.08) Diffuse: 1.22 (0.51, 2.93)		
					Proanthocyanidins	0.82 (0.71, 0.94) Male: 0.57 (0.36–0.93) Female: 1.22 (0.57–2.58) Cardia: 1.1 (0.52, 2.35) Noncardia: 0.84 (0.51, 1.38) Intestinal: 1.01 (0.5, 2.07) Diffuse: 0.87 (0.39, 1.98)		
Vitelli-Storelli et al. [5] Stop Project (Italy, Greece, Spain, Portugal, Mexico, Russia)	10 Case-control studies 1998–2015	Cases = 3471 Controls = 8344	FFQ	Vegetables, fruit, sweets, cereals, alcohol, juices and other drinks.	Total polyphenolsTotal flavonoids Anthocyanidins Flavanols FlavonolsFlavanones Total phenolic acids Hydroxybenzoic acids Hydroxycinnamic acids	0.67 (0.54–0.81) 0.73 (0.55–0.90) 0.74 (0.56–0.92) 0.77 (0.66–0.88) 0.76 (0.51–1.01) 0.57 (0.44–0.69) 0.75 (0.55–0.94) 0.73 (0.57–0.89) 0.82 (0.58–1.06)	Highest quartile X Lowest quartile	Age, sex, social class, alcohol consumption, BMI, family history of gastric cancer, smoking status, and salt consumption.
Rubín-García et al. [15] Spain	Multi-center Case-control 2008–2013	Cases = 329 Controls = 2700	FFQ past year	Legumes, vegetables, fruit, cereals, sweets and snacks, as well as alcoholic beverages and others.	Stilbenes Lignans Hydroxybenzaldehydes Hydroxycoumarin Tyrosols Other polyphenols	0.47 (0.32–0.69) 0.53 (0.36–0.84) 0.41 (0.28–0.61) 0.49 (0.34–0.71) 0.56 (0.39–0.80) 1.49 (1.06–2.10)	Highest quartile X Lowest quartile	Age; sex; socioeconomic status; smoking status; first-degree family history of GC; physical activity; BMI; alcohol consumption; and vegetables, red meat, salt, and total energy intake.
Kim et al. [32] Korea	Case-control 2011–2014	Cases = 415 Controls = 830	FFQ	Mostly fruit and vegetables	Total phenolics	Model 1/2 0.52 (0.37–0.75)/0.57 (0.39–0.83)	Highest tertile X Lowest tertile	Model 1: Age, BMI, education level, income, physical activity, smoking status, first-degree family history of GC, and total energy intake. Model 2: Model 1 + *H. pylori* infection status.

Note: Abbreviations: BMI—Body Mass Index; Cis—Confidence Intervals; FFQ—Food Frequency Questionnaire; GC—Gastric Cancer; GCA—Gastric Cardia Adenocarcinoma; OGA—Other Gastric Adenocarcinoma; ORs—Odds Ratios; RRs—Relative Risks. If there was log and quartile for a single variable, the “quartile” was chosen.

**Table 2 cancers-14-05878-t002:** Subgroup analyses of dietary polyphenol intake and gastric cancer risk.

Subgroup	Number of Participants	Number of Studies	RR (95% CI)	Heterogeneity Test	
				I^2^ (%)	*p*
All studies	1,197,857	19	0.71 (0.62–0.81) *	60.5	<0.001
Study design					
Cohort	1,171,647	7	0.88 (0.69–1.12)	54.7	0.039
Case–control	26,210	12	0.64 (0.56–0.74) *	44.4	0.049
Sex ¹					
Male	264,991	8	0.79 (0.67–0.94) *	31.6	0.176
Female	399,416	7	0.65 (0.48–0.87) *	49.7	0.064
Anatomical type ¹					
Cardia	1343	6	1.01 (0.79–1.27)	42.3	0.123
Noncardia	2603	6	0.85 (0.69–1.05)	65.2	0.013
Histological type ¹					
Diffuse	476	3	0.63 (0.37–1.09)	71.8	0.029
Intestinal	662	3	0.65 (0.52–0.82) *	0.0	0.494
Geographical area					
Asia	118,717	5	0.67 (0.51–0.89) *	60.7	0.038
America	470,985	3	0.87 (0.56–1.36)	78.3	0.010
Europe	608,155	11	0.67 (0.57–0.79) *	44.2	0.056
Adjustments					
Family history of gastric cancer, Yes	103,999	5	0.71 (0.59–0.86) *	59.3	0.044
No	1,093,858	14	0.70 (0.57–0.86) *	63.5	0.001
Fruit and/or vegetables intake ^2^, Yes	569,855	7	0.68 (0.55–0.83) *	49.5	0.065
No	638,597	16	0.72 (0.61–0.85) *	61.2	0.001

Note: ^1^ the total number of studies that analyzed the specific variable was considered. ^2^ the relative risk of the statistical models that used fruit and/or vegetable intake as an adjustment or otherwise was considered. * exhibited statistical significance.

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
