# Peer review of "Dietary Polyphenol Intake and Gastric Cancer: A Systematic Review and Meta-Analysis"

_cancers, 2022, doi:10.3390/cancers14235878_

Round 1

Reviewer 1 Report

Marcela de et al, in their paper discuss the potential of Dietary polyphenol intake on the pathogenesis / prognosis and treatment response among gastric cancer cohort. 

The study is quit impressive and cover a lot of aspect however in my own view, lack of immunological paradigm in patients who are fed with poly phenols is missing and should be included. 

This would give paper more visibility 

Reviewer 2 Report

  • The English language of the manuscript needs to be improved - there are grammatical mistakes.
  • Lines 14-17: You state that results are controversial for flavonoids - are there unknowns for other classes and polyphenol intake in total (the ones you explore) - if so, this needs to be stated here so that it is clear why this meta-analysis is done.
  • Line 23: Does population refer to geographical area?
  • Line 23: Consider being more precise regarding performing a subgroup analysis for adjustment variables. Namely, often studies provide only adjusted estimates and list various variables used for adjusting. Therefore, authors need to clarify how they performed a subgroup analysis by adjustment variables. In addition - for the main model, what kind of estimates did you extract and pool? The ones adjusted for most variables or? All of this needs to be briefly mentioned here and further explained in the section Methods.
  • Line 25: If this was a time frame you aimed for, this must be mentioned above in the methods subsection.
  • Lines 25-32: Why is heterogeneity not reported for all outcomes?
  • Lines 51-53: Are the authors sure about this? There was a paper published in Cancers which used individual participant data to pool results for total polyphenol intake and risk of gastric cancer - Vitelli-Storelli F, Rossi M, Pelucchi C, Rota M, Palli D, Ferraroni M, Lunet N, Morais S, López-Carrillo L, Zaridze DG, Maximovich D. Polyphenol intake and gastric cancer risk: Findings from the stomach cancer pooling project (stop). Cancers. 2020 Oct 20;12(10):3064.
  • Lines 59-60: Is this correct? Did you assess and meta-analyze different levels of intake?
  • Lines 70: What is meant by "predicted cancers"?
  • Lines 68-70: When presenting data on incidence and mortality, the year must be stated.
  • Line 83: As interesting as it is to provide a figure depicting the structure and classification of polyphenols, it seems even more relevant to provide a figure depicting their role in the carcinogenesis of gastric cancer - in line with the topic of the manuscript.
  • Lines 92-93: The sample size is not the only reason why results of different studies are not consistent - please provide a more comprehensive explanation here.
  • Lines 109-110: This is not clear enough. Seems more logical to state that articles were included if they assessed the association between the intake of dietary polyphenols and risk of GC.
  • Lines 119-120: If listing the third reason for exclusion - then you must state was considered for inclusion regarding how polyphenol intake was assessed and verified in participants.
  • Lines 126-127: Which OR/RR was extracted? The one most adjusted for or? Since you mention a subgroup analysis by these adjustment variables.
  • Lines 129-130: Explain transformations or assumptions made to combine ORs and RRs.
  • Line 131: Explain how heterogeneity was interpreted - e.g. cutoff values for I-squared, etc.
  • Lines 132-136: These sentences need to be clarified - what does it mean that the choice of ORs for analysis was done for..? And choice preference? What does it mean and how were these choices made?
  • Line 139: This subgroup analysis by adjustments needs to be explained here. What was extracted from a study? A risk estimate adjusted for a certain variable or not? What when studies usually provided only fully adjusted results (for multiple variables)? Clarify what was extracted and how it was combined.
  • Lines 141-142: Briefly explain what the NOS scale examines and that different scales are used for different study designs.
  • Lines 160-167: It is unusual to report percentages when describing how many studies were cohort and case-control studies, or characteristics as given in this paragraph - please provide only absolute numbers.
  • Lines 165-166: When you say - of these studies, it seems as if it relates to the previous sentence. Revise the English language here.
  • Line 165: 12 out of 19 is not most, it can be more than a half or similar.
  • Lines 167-168: And the other 3 used what? Provide complete information.
  • Lines 168-170: Were all studies adjusted for the same variables or were there differences? The writing needs to be precise and correct, if all studies were not adjusted for all of these variables then rather rephrase or just state that the Table shows this information.
  • Table 1: The methods section does not mention HR at all - this needs to be revised. Again, explanations must be provided for how OR/RR/HR were handled and pooled.
  • Table 1: The column for RR/OR/HR should be revised. As it currently stands it looks overcrowded and information is not clearly passed to the reader. Consider making some changes - omitting some of this information and using a format this detailed in the supplement. Here, decide which model you want to present - or for which polyphenol. Maybe present only statistically significant result or such.
  • Table 1: Methods section needs to describe that you assessed highest vs lowest intake when assessing the exposure - methods must define exposure and outcome.
  • Lines 184-185: Heterogeneity was high in many of the pooled estimates provided here. This sentence is misleading - as if a symmetric plot indicates that the found heterogeneity is not high/substantial. Not to mention that a funnel plot must be made for each separate outcome. Keep the description of publication bias separate. Heterogeneity must be acknowledged. Attempts to investigate it are through e.g. the subgroup analyses - this all must be clearly and transparently recognized by the authors.
  • Line 242: Seems that there could be other reasons which could explain the found differences by geography - outside of the polyphenols - these populations differ in other characteristics too, as well as the studies themselves surely.
  • Line 267: Cohort studies had a shorter follow-up than case-control studies? This is incorrect. The study design of case-control studies does not involve follow-up - the persons already have or do not have GC, while the included cohorts could be either prospective or retrospective, and if prospective then they had a follow-up, which however long is still longer than no follow-up in case-control studies. This must be corrected and reasons for differences by study design need to be properly addressed.
  • Lines 279-290: Other than an overview of what other studies found, either same or different, authors need to make an attempt to explain possible reasons for these findings.
  • Lines 304-311: Be specific when describing strengths of this study. What was found - a significant association for example, has nothing to do with strengths of this study, since not only positive findings are relevant. Further on, when trying to explain results of some subgroup analyses you state small sample sizes, here you state that as a strength that the sample size was large. Be clear about this throughout the manuscript. And be precise when stating - "we had a large sample size". This was not an individual participant data meta-analysis, which is also a limitation.
  • Lines 320-322: Lack of studies or what you tried to state here has nothing to do with limitations of your study - you already stated that it was your strength that others did not look at this problem the way you did. Limitations concern your analyses, heterogeneity, pooling choices, sample size, inherent limitations of the study design of included studies etc.
  • Were there any differences in what was done compared to what was planned in the PROSPERO registered meta-analysis protocol?

Round 2

Reviewer 2 Report

I would like to thank the Authors for their efforts in addressing the comments I have raised. Still, some issues remain:
  • Lines 24-25: I see that this sentence was revised, however the issue still remains. Specifically, instead of adjustment variables, state that you performed the analysis by - and then list the variables. Why? Because often a study provides an estimate that is fully adjusted and therefore you cannot perform a subgroup analysis by that variable used for adjustment. Saying that you used adjustment variables is incorrect, so just omit that word.
  • Regarding the response to Point 14 - would it be correct to specify this as "validated questionnaires"?
  • Regarding the response to Point 16 - this point was not addressed, and the response to previous point has nothing to do with this point. The question I raised was the combination of different risk measures used as risk estimates throughout the included studies - some studies reported OR some RR etc. And when you pool then in a meta-analysis you report one of those, and to do that you have to combine e.g. ORs and RRs. This requires certain assumptions or transformations to be done - please specify what was done.
  • Regarding the response to Point 19 - this also goes for all comments regarding the adjustment variables used for subgroup analysis. When a risk estimate is adjusted for a certain variable, that does not mean that you can take that estimate as the value for that variable and run a subgroup analysis by that variable - it means quite the opposite, that effect of that exact variable is removed from the observed relationship. This definitely needs to be rephrased everywhere in the manuscript and clarified - what was extracted and pooled when you state that you ran analyses by adjustment variables?
  • Regarding the response to point 31 - case-control studies have no follow up. So, do not mention follow-up at all when referring to case-control studies, their design means that all events already happened. I underline, no follow-up in case-control studies, so revise this.

Round 3

Reviewer 2 Report

I would like to thank the Authors for their efforts to improve the manuscript and to answer my comments and clarify all remarks. I believe that thanks to the changes the Authors have made, the manuscript has been significantly and comprehensively improved. Please see below one remaining comment:   - In regard to remark No. 3: Cite the two meta-analyses you mentioned here for this new sentence you added to the manuscript in Lines 132-134.